

# Virtual reality therapy in managing cancer pain in middle-aged and elderly: a systematic review and meta-analysis

Yang Chen[1,*], Hui Meng[2,*], Qian Chen[3], Wendong Wu[4], HaiBin Liu[5], Shi Lv[6] and Liang Huai[5]

[1] Department of Rehabilitation, Taian Maternal and Child Health Hospital, Taian, China
[2] Department of Joint and Sports Medicine, Second Affiliated Hospital of Shandong First Medical University, Taian, China
[3] Department of Rehabilitation, Second Affiliated Hospital of Shandong First Medical University, Taian, China
[4] Department of Rehabilitation, 88 Hospital, Taian, China
[5] School of Sports Medicine and Rehabilitation, Shandong First Medical University, Taian, China
[6] Institute of Brain Science and Brain-inspired Research & Department of Neurology, Second Affiliated Hospital of Shandong First Medical University, Taian, China
* These authors contributed equally to this work.

Corresponding authors
Shi Lv, ls15666080332@163.com
Liang Huai, hlniren@163.com

## ABSTRACT

**Background:** Virtual reality technology has been proposed to rehabilitate cancer patients. This study aimed to summarize the effectiveness of virtual reality (VR)–based therapies for pain management in middle-aged and elderly cancer patients.
**Methods:** This meta-analysis was registered in PROSPERO (CRD42023400432). We searched the randomized controlled trials (RCTs) in PubMed, Scopus, the Cochrane Library, Web of Science, and Embase, conducted from construction until November 1, 2024. The study examined the effects of VR treatment on pain levels in middle-aged and elderly cancer patients using RCTs as primary or secondary outcome measures. Articles were evaluated for eligibility according to predetermined criteria, and each of the three researchers independently collected the data. The researchers used the heterogeneous selection effects model to calculate the mean effect sizes.
**Results:** This meta-analysis included seven RCTs involving 476 patients. The meta-analysis confirmed the significant effect of VR therapy on the management of pain, anxiety, and depression in the middle-aged and elderly cancer population.
**Conclusions:** Our research shows that VR could be a significant device for cancer pain management in the middle-aged and elderly and that VR scene therapy may be more effective. Nevertheless, it is essential to use caution when interpreting the findings since the number of research included is small.

## INTRODUCTION

Tumors seem to be especially significant today because of their substantial adverse health, psychosocial, and economic impacts. Because cancer is becoming more commonplace globally, it is expected to overtake all other causes of death by 2030 (*Mattiuzzi & Lippi, 2019*).

Therefore, actions to improve the present cancer patient rehabilitation techniques are required. Modern designs are increasingly employed in addition to conventional cancer therapies (*Bishayee & Block, 2015*). Due to the increased availability of targeted and chemotherapeutic agents brought about by new mechanisms of action, the effectiveness of cancer treatments has improved (*Poort et al., 2020*). Nevertheless, this has also amplified the likelihood of experiencing weariness, which is interrelated to diminished psychological anguish and worse physical performance. Middle-aged and elderly individuals often experience pain and anxiety, and these sensations may be intensified as a result of medical procedures and the adverse effects of drugs during the treatment of cancer (*Chow et al., 2021*). Pain and anxiety are common adverse effects experienced by middle-aged and elderly individuals undergoing cancer treatment, mainly as a result of chemotherapy and medical interventions. Factors such as hospitalisation for diagnosis and treatment, separation from family, and dread of mortality may also contribute to anxiety (*Fitzmaurice et al., 2017*). Anxiety and pain are two sensations that may have a substantial impact on the quality of cancer therapy in middle-aged and elderly populations. Hence, it is essential to implement distinct and focused therapeutic interventions to effectively manage anxiety and discomfort in the middle-aged and elderly population receiving cancer treatment. Meanwhile, with non-pharmacological therapies like music, acupuncture, *etc.*, the advancement of technology is leading to the emergence of novel therapeutic methods that are becoming more popular (*Satija & Bhatnagar, 2017*).

Virtual reality (VR) technology allows users of computer simulation systems to explore virtual worlds (*Garcia-Palacios et al., 2015*). By using computers to create a simulated environment where users can interact, virtual reality creates an immersive experience that makes it easier for users to feel physically present in the VR world (*Hoffman et al., 2004*). In the last several years, virtual reality has gained more traction and acceptance due to the introduction of more affordable gadgets like head-mounted displays (*Hoffman et al., 2006*). Contrary to most pain relievers, which interfere with the C-fiber route responsible for transmitting harmful signals to the central nervous system, VR alters the impression of pain by modifying attention, concentration, and emotions (*Wismeijer & Vingerhoets, 2005*). VR creates immersive environments that reduce pain perception by upregulating non-painful brain impulses (*Sharar et al., 2007*). As an additional therapeutic alternative for burn pain, acute pain, and experimental pain management in adults, virtual reality is gaining more and more acceptance (*Carrougher et al., 2009*; *Hoffman et al., 2007*). Research has shown that VR lessens pain during various medical procedures, including chemotherapy and wound care, and it has also been proven beneficial when used in conjunction with medications (*Gutiérrez-Maldonado et al., 2012*). Individuals experience VR's virtual world, giving them a realistic feeling of being there. The level of presence is determined by the VR aspects displayed to the user. Individuals may develop a psychologically solid belief that they are in the virtual realm rather than the real world, where pain is experienced. This can occur when they can visually perceive parts of their body in the virtual environment and effectively ignore external distractions from the actual world (*Persky & Lewis, 2019*). Individuals may have a strong sense of presence that

diminishes their susceptibility to unpleasant stimuli and detrimental brain impulses, alleviating their pain perception (*Pourmand et al., 2018*). The use of distraction has characterised this procedure. Distraction analgesia (*Hoffman et al., 2007*) is the primary mechanism associated with the pain-relieving benefits of virtual reality. It is the basis for all pain management methods employed in this setting. Distraction assists by immersing the patient in a virtual environment and redirecting their attention away from the unpleasant stimuli. The neural substrate theory of pain posits that alterations in pain output may be attributed to factors that affect the inputs, such as attention and inputs themselves, including emotion, sensation, and cognition. This idea forms the basis of distraction therapy. This process, associated with pain sensation, is similar to several analgesics that block the pathways responsible for transmitting nociceptive signals to the CNS (*Pourmand et al., 2018*). There is a finite amount of attentional resources available in cognitive processes, and when there are sensory distractions, fewer resources are allocated to pain processing. Consequently, it is hypothesized that integrating many sensory modalities reduces the chances for individuals to experience pain (*Pancekauskaitė & Jankauskaitė, 2018*). Additionally, VR analgesia can be achieved by redirecting attention and enhancing skill development.

Existing research suggests that VR interventions may be a beneficial method of diversion for chemotherapy patients for various types of cancer (*Sajeev et al., 2021*; *Schneider & Hood, 2007*; *Schneider, Kisby & Flint, 2011*). However, there is a scarcity of meta-analyses examining the effectiveness of VR interventions in alleviating cancer pain in middle-aged and elderly patients during chemotherapy. This meta-analysis and systematic review aimed to offer practical guidance on implementing novel alternative approaches for managing cancer pain and assess virtual reality's effectiveness in alleviating cancer pain in middle-aged and elderly individuals.

# MATERIALS AND METHODS

## Registration program

This meta-analysis was registered in PROSPERO (CRD42023400432). The meta-analysis employed the fundamental components of the PRISMA-P for conducting a systematic assessment (*Moher et al., 2009*).

## Search strategy

Three researchers (Y.C, H.M, Q.C) made comprehensive inquiries in the PubMed, Scopus, Cochrane Library, Web of Science, and Embase databases using title, abstract, and keyword terms. The search lasted from inception until December 1, 2023. PubMed, Cochrane Library and Embase databases Search Strategies were listed in the Supplemental Appendix. Ultimately, the assessor thoroughly examined the list of references to be included in the research. There were no restrictions or constraints on the searches. Two investigators positively assessed the research to see whether it fulfilled the criteria for inclusion. When the two investigators could not reach a consensus, they spoke with a third investigator (L.H.) to decide whether to accept or reject the research. The searches included
the following keywords: "virtual reality," "VR," "virtual reality therapy," "rehabilitation," "cancer," "tumour," "psychological," "symptom," "pain," "anxiety," and "depression".

## Inclusion and exclusion criteria

The research's inclusion criteria were determined using the PICOS methodology (*Cumpston et al., 2019*). P (population): a middle-aged and elderly (median age ≥ 50) population with cancer diagnoses, comprising survivors, patients receiving active therapy, and patients diagnosed but not yet starting treatment; I (intervention): any type of VR-based training, including game-based training, semi-immersive training, and immersive training, regardless of the application, environment, duration, or number of sessions; C (control): no VR intervention or standard treatment (such as regular rehabilitation, health education, or psychological care); O (outcome): to ascertain how a virtual reality intervention affects cancer patients' pain, anxiety, and quality of life; S (design): RCTs released in English before November 1, 2024. The exclusion criteria are uncertain interventions, ambiguous case data, duplicate publications, non-randomized controlled trials, and non-English research.

## Extract data

Each of the three assessors created a data extraction form. After the agreement, the table was incorporated into the final form's standardization. Including (1) research ID (author, Publication year) and study location (country/region). (2) Key study population variables include demographic factors such as age and sex, the sample sizes of the experimental and control group, and the specific kind of cancer under investigation. (3) Group characteristics include the type of intervention, the setting, the dosage, and the length of therapy. (4) Evaluation indicators based on assessment scales. (5) A review of effectiveness based on the main result. Three evaluators independently retrieved the primary outcome, compared the included research, and jointly analyzed the results. After disputes were settled by debate, the final table was created. We obtained the main result's average and standard deviation for a quantitative meta-analysis.

## Risk of bias assessment and quality assessment

The risk of research bias was evaluated from seven items (Table S1). Two researchers (Y.H. and Q.N.) closely adhered to the Cochrane Handbook's risk assessment tool (*Higgins et al., 2011*). As described in our previous study (*Chen et al., 2024*), The methodological quality of the study was evaluated using the PEDro scale (Table S2).

## Statistical analysis

The standardized mean difference (SMD) and mean difference (MD) with 95% confidence intervals are shown using RevMan 5.4 for meta-analysis of continuous data. SMD was computed by evaluating the same result using several measuring instruments (*Faraone, 2008*). The Cochran Q test and Higgins $I^2$ statistics were used to evaluate the heterogeneity of the study. A fixed-effects model was employed when heterogeneity was deemed modest, as shown by the values of $P$ and $I^2$ when $P > 0.1$ and $I^2 < 50\%$ (*Higgins & Thompson, 2002*).

A random-effects model was used to find the origins of heterogeneity, and subgroup analyses were carried out when heterogeneity was substantial ($P < 0.1$ and $I^2 > 50\%$). It was done utilizing the leave-one-out method for sensitivity analysis. It was deemed statistically significant when $P < 0.05$.

## RESULT

### Research inclusion

A predefined search strategy was used to get 432 distinct categories of research. Duplicate studies, non-RCTs, Inconsistent interventions, Inconsistent outcome indicators, and unretrievable data were all excluded. The meta-analysis includes seven articles selected by examining the abstracts and titles (Table 1 summarizes the age baseline of the participants in this study) (*Bani Mohammad & Ahmad, 2019*; *Basha et al., 2022*; *Feyzioğlu et al., 2020*; *Gao et al., 2022*; *Turrado et al., 2021*; *Villumsen et al., 2019*; *Zhang et al., 2022*). The study consisted of 476 participants diagnosed with cancer, with 234 assigned to the VR group and 242 to the conventional rehabilitation group. The selected research is shown in Fig. 1.

### Research characteristics

Seven studies have been conducted in six different countries. These studies provide information on various aspects, including study ID, study population, experimental and control group characteristics, primary outcome indicators, and evaluation effects. This information is presented in Table 1.

### Methodological quality

The Cochrane Handbook was used for evaluating the included research. Every research item was subjected to rigorous randomisation and control. The research underwent a thorough examination to identify any bias or selective publishing. The evaluation of bias is shown in Figs. 2 and 3.

## META-ANALYSIS RESULTS

### Pain

Four trials (*Bani Mohammad & Ahmad, 2019*; *Basha et al., 2022*; *Feyzioğlu et al., 2020*; *Villumsen et al., 2019*) assessed how well virtual reality treatment reduced cancer pain and involved 213 cancer patients. The findings demonstrated a significant difference between the two groups ($n = 4$; SMD = −3.76, 95% CI [−6.52 to −1.00], $P < 0.001$). They were using a random effects model to account for the substantial variability ($I^2 = 93\%$, $P < 0.0001$) (Fig. 4A). The VR scene was substantially more successful than the conventional rehabilitation group in reducing pain in the VR scene subgroup ($n = 2$; SMD = −2.89, 95% CI [−4.87 to −0.92], $P < 0.01$). However, in the VR game training subgroup, there was no discernible difference between the VR and control group ($n = 2$; SMD = −4.91, 95% CI [−14.01 to 4.18], $P > 0.05$). The groups did not show a statistically significant interaction ($\chi^2 = 0.18$, $P = 0.67$, $I^2 = 0\%$) (Fig. 4B).
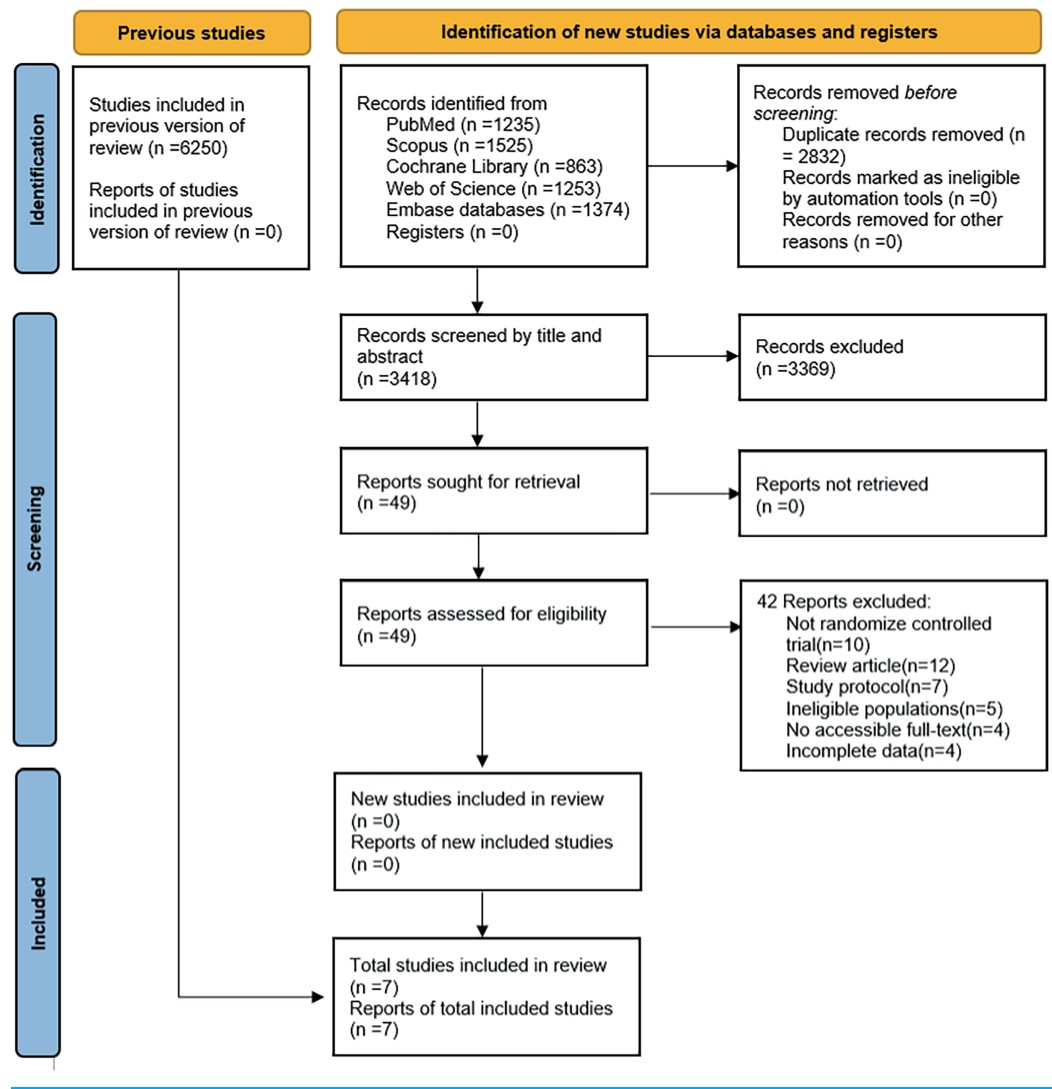

**Figure 1 Flow chart for research screening.**

## Anxious

Four studies (*Bani Mohammad & Ahmad, 2019*; *Gao et al., 2022*; *Turrado et al., 2021*; *Zhang et al., 2022*) examined the impact of VR therapies on anxiety in a total of 339 individuals diagnosed with cancer. A statistically significant difference was seen between the two groups ($n = 4$; SMD = −9.19. 95% CI [−14.23 to −4.16], $P < 0.001$). Due to the high heterogeneity of this outcome, Using a random effects model ($I^2 = 85\%$, $P < 0.001$) (Fig. 5).

## Depressed

Two studies (*Turrado et al., 2021*; *Zhang et al., 2022*), including 203 cancer patients, examined the impact of a VR intervention on depressed mood. The findings demonstrated a statistically significant disparity between the between the two groups ($n = 2$; SMD = −0.45, 95% CI [−0.73 to −0.17], $P < 0.01$), using a fixed-effects model ($I^2 = 0\%$) (Fig. 6).

Table 1 Key attributes of the research that were included.

| Author (year) country | Patients number (E/C) | Age (E/C) | Cancer type | Intervention | Intervention time | Evaluation indicators | Effectiveness evaluation | References (PMID) |
|---|---|---|---|---|---|---|---|---|
| Gao et al. (2022) China | 60 30/30 | 55.82 ± 11.98 | Chest tumor | Patients undergo a virtual experience of a radiation therapy (RT) procedure, a virtual patient demonstration of the use of radiation beams during treatment, and RT positioning and simulation. | 30 min/d | STAI | VR educational intervention reduces anxiety in cancer patients | 32829456 |
| Turrado et al. (2021) Spanish | 126 58/68 | 64 (41–85)/ 68 (50–86) | Colorectal cancer | Put on your VR glasses and experience the treatment in the VR APP. | (16-30) min/d | STAI HADS | VR therapy in clinical care settings can improve patient health. | 33683433 |
| Feyzioğlu et al. (2020) Istanbul | 36 19/17 | 55.84 ± 8.53/ 56.00 ± 7.06 | Lymphoma | Kinect-based VR rehabilitation program where patients perform activities such as darts, bowling, and boxing in a virtual environment. | 45 min/3d | VAS | Kinetics-based VR therapy produced significant results comparable to standard physical therapy in the early postoperative period. | 31907649 |
| Basha et al. (2022) Saudi Arabia | 60 30/30 | 58.83 ± 7.0/ 59.07 ± 7.48 | Lymphoma | Includes "Macarena" dance and other Xbox Kinect games for VR rehab. | This was done 5 times a week for 8 weeks. | VAS | VR is an innovative and effective solution that can enhance physical functionality and overall quality of life with breast cancer. | 34669036 |
| Mohammad & Ahmad (2018) Jordan | 76 38/38 | 57.25 ± 6.32 | Lymphoma | VR simulated scenarios for training or relaxing with a headset and headphones. | End before the peak of painkiller efficacy. | STAI, VAS | Immersive VR is more effective as an adjunctive intervention than morphine alone, and VR is a safer intervention than medication. | |
| Villumsen et al. (2019) Denmark | 41 21/20 | 67.6 ± 4.6/ 69.8 ± 4.4 | Prostate cancer | The individual utilizes a virtual reality head-mounted display and selects from three virtual simulation encounters encompassing simulated parks and tourism destinations. | Each session lasts about 1 h, thrice a week for 12 weeks. | FACT-P; VAS; QOL | VR interventions appear to be safe and can improve the health of the prostate cancer patient population. | 31012238 |
| Zhang et al. (2022) China | 77 38/39 | 57.29 ± 7.69/ 59.03 ± 7.98) | Lymphoma | Wearing a VR device for an audio-visual fusion experience constitutes a complete mode of intervention. | Six sessions of 30 min each over 3 months | SAS; QOL | VR has a beneficial effect on improving the quality of life by alleviating psychological distress, anxiety, and other multifaceted disorders in breast cancer patients. | 35712124 |

Note:
E/C, Experimental group/control group; STAI, State-trait anxiety scale; HADS, Hospital anxiety and depression scale; VAS, Visual analog scale; QoL, Quality of life; FACT-P, Functional assessment of cancer therapy–prostate; SAS, Self-Rating anxiety scale.
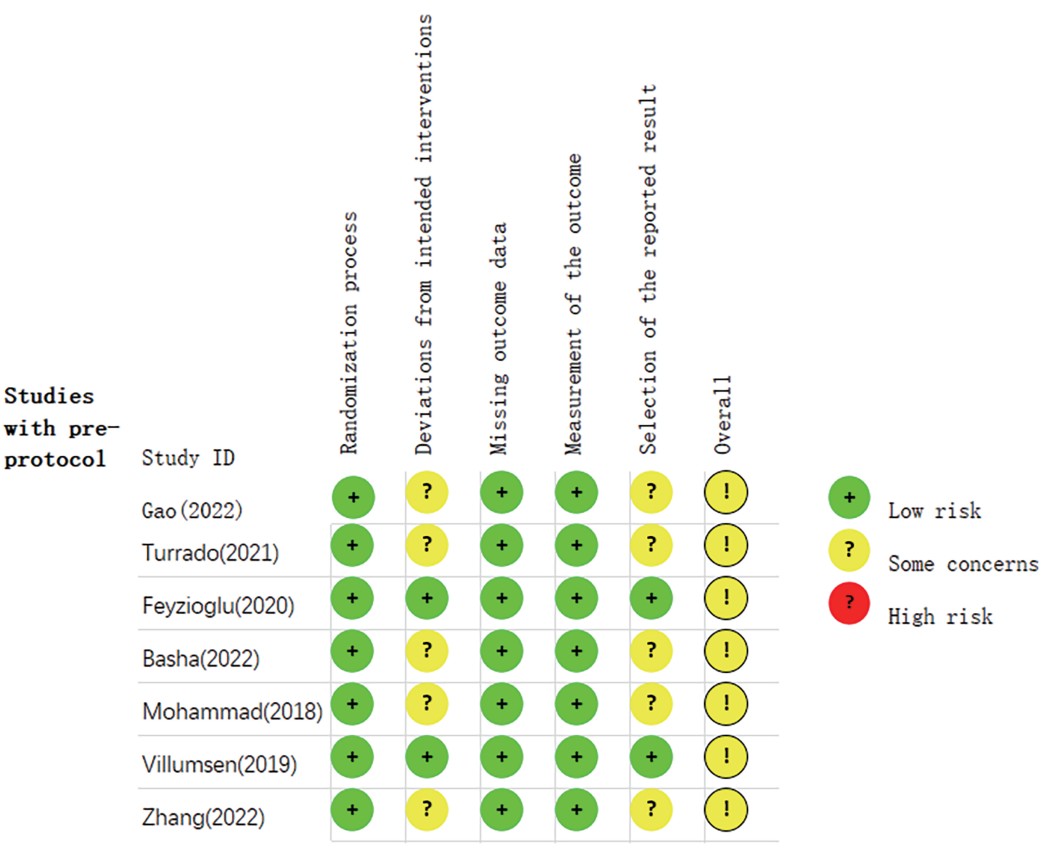

**Figure 2** Summary graph of risk of bias of research (*Gao et al., 2022*; *Turrado et al., 2021*; *Feyzioğlu et al., 2020*; *Basha et al., 2022*; *Mohammad & Ahmad, 2018*; *Villumsen et al., 2019*; *Zhang et al., 2022*).

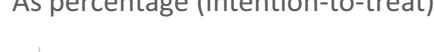

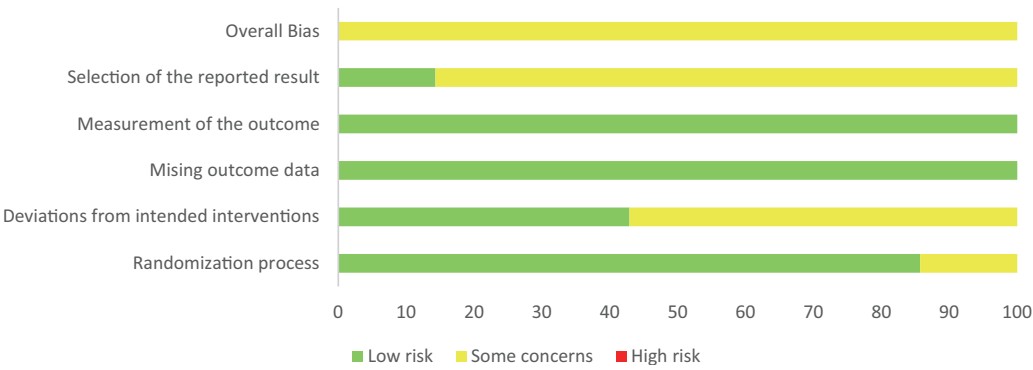

**Figure 3** The risk assessment of biased dependence of research.

## Quality of life

Two studies (*Villumsen et al., 2019*; *Zhang et al., 2022*) involving 118 cancer patients examined the VR therapies on the patient's quality of life (QOL). There was no statistically

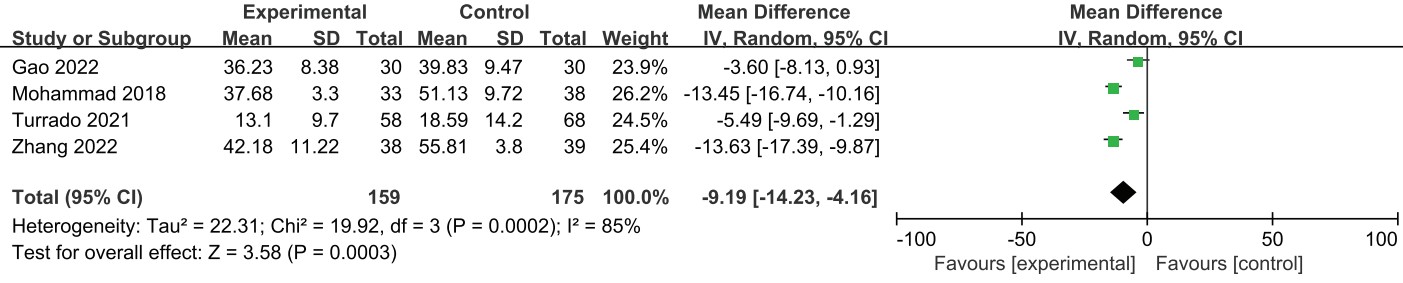

**A**

| Study or Subgroup | Experimental Mean | SD | Total | Control Mean | SD | Total | Weight | Mean Difference IV, Random, 95% CI |
|---|---|---|---|---|---|---|---|---|
| Basha 2022 | 41.23 | 10.3 | 30 | 51.17 | 10.45 | 30 | 14.5% | -9.94 [-15.19, -4.69] |
| Feyzioglu 2020 | 1.53 | 1.3 | 19 | 2.16 | 1.72 | 17 | 29.3% | -0.63 [-1.64, 0.38] |
| Mohammad 2018 | 0.33 | 0.72 | 38 | 4.94 | 2.27 | 38 | 29.8% | -4.61 [-5.37, -3.85] |
| Villumsen 2019 | 2.34 | 3.22 | 21 | 5.22 | 3.25 | 20 | 26.4% | -2.88 [-4.86, -0.90] |
| **Total (95% CI)** | | | **108** | | | **105** | **100.0%** | **-3.76 [-6.52, -1.00]** |

Heterogeneity: Tau² = 6.51; Chi² = 44.86, df = 3 (P < 0.00001); I² = 93%
Test for overall effect: Z = 2.67 (P = 0.008)

**B**

| Study or Subgroup | Experimental Mean | SD | Total | Control Mean | SD | Total | Weight | Mean Difference IV, Random, 95% CI |
|---|---|---|---|---|---|---|---|---|
| **1.1.1 Virtual scene** | | | | | | | | |
| Mohammad 2018 | 0.33 | 71 | 38 | 4.94 | 2.27 | 38 | 2.0% | -4.61 [-27.20, 17.98] |
| Villumsen 2019 | 2.34 | 3.22 | 21 | 5.22 | 3.25 | 20 | 36.8% | -2.88 [-4.86, -0.90] |
| **Subtotal (95% CI)** | | | **59** | | | **58** | **38.8%** | **-2.89 [-4.87, -0.92]** |

Heterogeneity: Tau² = 0.00; Chi² = 0.02, df = 1 (P = 0.88); I² = 0%
Test for overall effect: Z = 2.87 (P = 0.004)

| | | | | | | | | |
|---|---|---|---|---|---|---|---|---|
| **1.1.2 Virtual game training** | | | | | | | | |
| Basha 2022 | 41.23 | 10.3 | 30 | 51.17 | 10.45 | 30 | 20.2% | -9.94 [-15.19, -4.69] |
| Feyzioglu 2020 | 1.53 | 1.3 | 19 | 2.16 | 1.72 | 17 | 41.0% | -0.63 [-1.64, 0.38] |
| **Subtotal (95% CI)** | | | **49** | | | **47** | **61.2%** | **-4.91 [-14.01, 4.18]** |

Heterogeneity: Tau² = 39.62; Chi² = 11.65, df = 1 (P = 0.0006); I² = 91%
Test for overall effect: Z = 1.06 (P = 0.29)

| **Total (95% CI)** | | | **108** | | | **105** | **100.0%** | **-3.42 [-6.68, -0.16]** |

Heterogeneity: Tau² = 6.47; Chi² = 14.62, df = 3 (P = 0.002); I² = 79%
Test for overall effect: Z = 2.06 (P = 0.04)
Test for subgroup differences: Chi² = 0.18, df = 1 (P = 0.67), I² = 0%

**Figure 4** (A) Forest plot demonstrates VR's effect on the emotional state of middle-aged and elderly cancer patients experiencing pain. (B) Subgroup analysis of the effect of two VR therapies (*Basha et al., 2022*; *Feyzioğlu et al., 2020*; *Mohammad & Ahmad, 2018*; *Villumsen et al., 2019*).

| Study or Subgroup | Experimental Mean | SD | Total | Control Mean | SD | Total | Weight | Mean Difference IV, Random, 95% CI |
|---|---|---|---|---|---|---|---|---|
| Gao 2022 | 36.23 | 8.38 | 30 | 39.83 | 9.47 | 30 | 23.9% | -3.60 [-8.13, 0.93] |
| Mohammad 2018 | 37.68 | 3.3 | 33 | 51.13 | 9.72 | 38 | 26.2% | -13.45 [-16.74, -10.16] |
| Turrado 2021 | 13.1 | 9.7 | 58 | 18.59 | 14.2 | 68 | 24.5% | -5.49 [-9.69, -1.29] |
| Zhang 2022 | 42.18 | 11.22 | 38 | 55.81 | 3.8 | 39 | 25.4% | -13.63 [-17.39, -9.87] |
| **Total (95% CI)** | | | **159** | | | **175** | **100.0%** | **-9.19 [-14.23, -4.16]** |

Heterogeneity: Tau² = 22.31; Chi² = 19.92, df = 3 (P = 0.0002); I² = 85%
Test for overall effect: Z = 3.58 (P = 0.0003)

**Figure 5** A forest plot illustrating the impact of VR on anxiety levels among middle-aged and elderly individuals diagnosed with cancer (*Gao et al., 2022*; *Turrado et al., 2021*; *Mohammad & Ahmad, 2018*; *Zhang et al., 2022*).

significant difference between the two groups ($n = 2$; SMD = 16.50, 95% CI [−12.95 to 45.95], $P > 0.05$) using a fixed-effect model ($I^2 = 97\%$) (Fig. 7).

## Pain in all age groups

We included supplementary research to investigate the effects of VR game training on pain in cancer patients. Four studies were conducted with middle-aged and elderly individuals

| Study or Subgroup | Experimental Mean | SD | Total | Control Mean | SD | Total | Weight | Std. Mean Difference IV, Fixed, 95% CI |
|---|---|---|---|---|---|---|---|---|
| Turrado 2021 | 2.2 | 3.8 | 58 | 5.1 | 8.26 | 68 | 62.1% | -0.44 [-0.79, -0.08] |
| Zhang 2022 | 46.63 | 9.82 | 38 | 50.21 | 3.8 | 39 | 37.9% | -0.48 [-0.93, -0.02] |
| **Total (95% CI)** | | | 96 | | | 107 | 100.0% | **-0.45 [-0.73, -0.17]** |

Heterogeneity: Chi² = 0.02, df = 1 (P = 0.89); I² = 0%
Test for overall effect: Z = 3.18 (P = 0.001)

**Figure 6 Forest plot of the effect of VR on depressed mood in middle-aged and elderly cancer patients (*Turrado et al., 2021*; *Zhang et al., 2022*).**

| Study or Subgroup | Experimental Mean | SD | Total | Control Mean | SD | Total | Weight | Mean Difference IV, Random, 95% CI |
|---|---|---|---|---|---|---|---|---|
| Villumsen 2019 | 121.9 | 13.6 | 21 | 120.7 | 16.2 | 20 | 49.1% | 1.20 [-7.98, 10.38] |
| Zhang 2022 | 104.7 | 13.33 | 38 | 73.44 | 6.5 | 39 | 50.9% | 31.26 [26.56, 35.96] |
| **Total (95% CI)** | | | 59 | | | 59 | 100.0% | **16.50 [-12.95, 45.95]** |

Heterogeneity: Tau² = 437.96; Chi² = 32.63, df = 1 (P < 0.00001); I² = 97%
Test for overall effect: Z = 1.10 (P = 0.27)

**Figure 7 Forest plot of the effect of VR on quality of life (*Villumsen et al., 2019*; *Zhang et al., 2022*).**

(*Bani Mohammad & Ahmad, 2019*; *Basha et al., 2022*; *Feyzioğlu et al., 2020*; *Villumsen et al., 2019*) and five with adolescents (*Gerçeker et al., 2021*; *Gershon et al., 2004*; *Tennant et al., 2020*; *Wolitzky et al., 2005*; *Wong et al., 2021*). These studies were utilised to evaluate the influence of a VR-based intervention on cancer-related pain. The results revealed a statistically significant difference between the two groups ($n = 9$; SMD = −12.95, 95% CI [−4.06 to −1.84], $P < 0.001$). using a random effects model ($I^2 = 83\%$, $P < 0.001$) (Fig. 8A).

The study found that in the subgroup of VR type, VR scenes were more effective than the control group in reducing pain ($n = 6$; SMD = −3.30, 95% CI [−4.18 to −2.42], $P < 0.01$). Additionally, in the subgroup of VR game training, interventions based on VR game training showed a significant improvement in pain compared to the control group ($n = 3$; SMD = −2.68, 95% CI [−5.08 to −0.29], $P < 0.05$). The analysis revealed no statistically significant association between subgroups ($\chi^2 = 0.22$, $P = 0.64$, $I^2 = 0\%$) (Fig. 8B). The virtual reality intervention resulted in substantial reductions in pain reported by middle-aged and elderly ($n = 4$; SMD = −3.76, 95% CI [−6.52 to −1.00], $P = 0.008$) and teenage ($n = 5$; SMD = −2.70, 95% CI [−3.25 to −2.16], $P < 0.001$) cancer patients compared to the control group. The subgroup interaction did not yield statistically significant results ($\chi^2 = 0.54$, $P = 0.46$, $I^2 = 0\%$) (Fig. 8C). The meta-analysis's findings demonstrated that both virtual reality-based therapies significantly reduced cancer patients' perceptions of pain, with VR scenes having a more significant impact than VR games. We discovered by comparing them across the age group that this impact is not only beneficial for managing pain in middle-aged and older individuals but also for teenage cancer patients. As a result, it may be used to treat cancer patients of all ages.

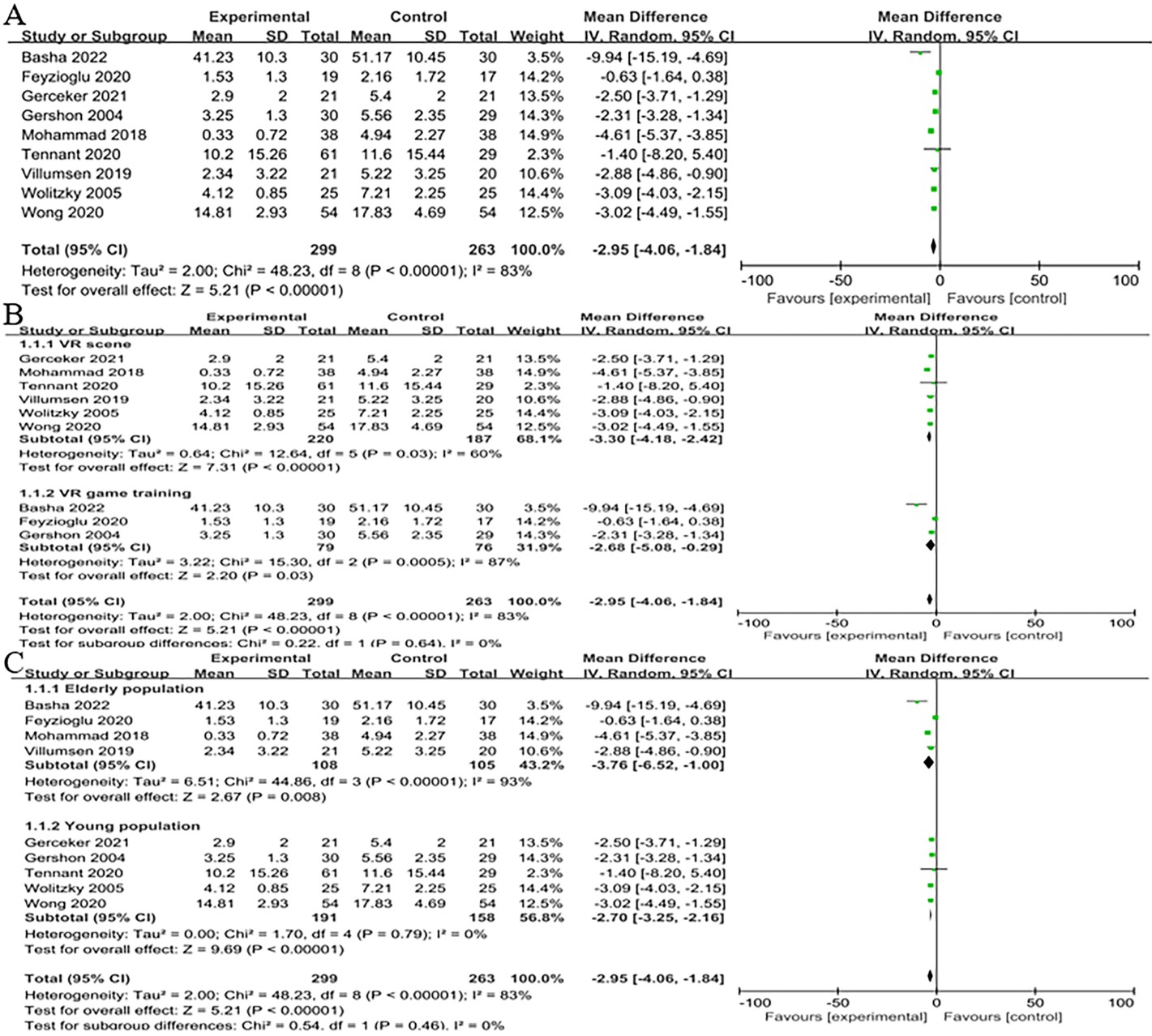

**Figure 8** (A) Forest plot of the effect of VR on pain across the age group. (B) Subgroup analysis was conducted to assess the impact of various virtual reality interventions on pain and attitude in cancer patients of different age groups. (C) Subgroup analysis to examine the effects of VR on pain across patients across various age groups (*Basha et al., 2022*; *Feyzioğlu et al., 2020*; *Gerçeker et al., 2021*; *Gershon et al., 2004*; *Mohammad & Ahmad, 2018*; *Tennant et al., 2020*; *Villumsen et al., 2019*; *Wolitzky et al., 2005*; *Wong et al., 2021*).

## Sensitivity analysis

The sensitivity analysis results indicated that excluding individual trials one at a time had no discernible impact on the overall results. This implies the meta-analysis's results are solid (Figs. S1–S3).

## Publication bias assessment

When fewer than ten articles are included in a meta-analysis of outcome measures, funnel plots are typically not suggested for publication bias analysis (*Chang et al., 2015*). First, Publication bias was a significant concern due to the small sample size of RCTs (*Egger et al., 1997*). Second, there is Heterogeneity in the type of cancer, chemotherapy regimen, and rehabilitation program design of patients in RCTs. Third, the inclusion of English-only RCTs may affect publication bias.

## DISCUSSION

Our research aims to analyze the effectiveness of a VR for middle-aged and elderly cancer patients. This is the first time such an evaluation has been undertaken. The results indicated that VR therapies had a statistically significant effect on decreasing pain, anxiety, and depression levels among middle-aged and elderly cancer patients. Despite the lack of a notable disparity in cancer pain alleviation between the VR game training group and the traditional rehabilitation group, we accounted for the diversity and sizes of the samples by forming subgroups for VR scene and VR game training. Additionally, many additional studies examining VR on pain in teenagers with cancer were also included. The meta-analysis findings indicate that both VR-based therapy approaches have a substantial impact on reducing pain perception in cancer patients. Although our findings deviated from the results reported by *Zeng et al. (2019)*, their study ultimately determined that VR-based treatment did not have a discernible effect on the pain, anxiety, or despair experienced by cancer patients. The variation in outcomes between the two studies might be attributed to the diverse array of virtual reality interventions, including differences in amount, quality, and kind used in the included trials. Our research indicates that VR therapies might potentially replace traditional therapy for treating mental health issues in middle-aged and elderly cancer patients. The quality of life of cancer patients is a multifaceted factor that is influenced by several circumstances, such as the kind of disease, its progression, and the treatment administered. Enhancing the quality of life for patients may be achieved by facilitating psychosocial issues connected with cancer. Our quantitative studies revealed that virtual reality therapies did not substantially impact the quality of life of middle-aged and elderly cancer patients.

These studies focused on two types of VR-based therapeutic interventions: VR game training and VR scenes. Studies have shown that using VR scenes can help cancer patients undergoing chemotherapy feel less depressed, anxious, painful, and exhausted (*Bani Mohammad & Ahmad, 2019*; *Chirico et al., 2020*; *Chow et al., 2021*). According to these studies, cancer patients find solace in the natural environment similar to their living conditions. This aligns with their instinctive belief that natural settings promote psychological health, prosperity, and survival. Patients can temporarily leave the hospital setting, interact with the outside world, and lose themselves in a soothing environment that increases their chances of physical and psychological healing and lessens unpleasant feelings (*Zhang et al., 2022*). This is made possible by virtual reality therapy's sense of presence. This process has been called distraction, and strategies for utilizing VR to reduce pain often rely on how distracted one is (*Pal, Cortiella & Herndon, 1997*). Distracting

stimuli can reduce pain perception because they downregulate damaging brain impulses. Instead of blocking the C-fiber pathway that causes negative signals to reach the central nervous system, as many analgesics do, VR changes pain perception by affecting attention, focus, and mood (*Sharar et al., 2007*). The immersive VR environment reduces the experience of pain by upregulating brain impulses that are not connected to pain. VR may also lower heart rate, anxiety, and the unpleasantness of pain, among other pain-related parameters. Fear of pain (or concern related to pain) may cause subjective pain to increase (*Arntz, Dreessen & De Jong, 1994*), with effects that seem to be mediated by attentional focus, whereby diverting attention away from pain reduces subjective pain ratings. One study (*Janssen & Arntz, 1996*) found that although fear of pain led engagement to be distracted from painful stimuli, increasing emotional discomfort, other distracting factors prevented pain from being felt as much. Game-based rehabilitation training enhances early activity compared to traditional rehabilitation training using a simulation network that permits patients to interact with a VR environment. This is achieved by providing patients access to various sensory stimuli, including touch, sight, and sound, and enhancing their motivation and adherence to therapy (*Basha et al., 2022*; *Feyzioğlu et al., 2020*; *Wong et al., 2021*). The kind of exercise and recuperation period are vital areas where VR and traditional regimens diverge. The most effective strategies and programs are still up for grabs. In the future, developing a tailored virtual reality rehabilitation training program that considers the patient's degree of motor impairment would be beneficial. When adjusting the exercise intensity, it is essential to consider the extent of improvement in motor function (*Atef et al., 2020*). The results of our meta-analysis point to VR scene treatment as being more statistically significant than VR game training for lowering cancer pain in the senior population. However, we could not directly compare the two treatment approaches due to the sample size's limited inclusion.

VR has been widely utilized in several domains of chronic pain management, including cancer pain therapy when conventional medications are insufficient for addressing chronic pain. This is significant in the economic, social, and health implications at both national and global scales. The specific mechanisms behind the transition from acute to chronic pain are still not well understood (*Glare, Aubrey & Myles, 2019*). This leads to alterations in the heightened pain transmission system and the neurochemicals and brain connections associated with pain's sensory and emotional elements. Like prior non-pharmacological treatments, virtual reality can selectively impact different brain areas. To attain enduring pain management, individuals with persistent ailments must acquire assurance *via* acquiring enduring techniques that they may use without the constraints of a virtual reality headset. Indeed, some individuals may refrain from engaging in virtual reality experiences due to the apprehension that it may elicit discomfort (*O'Sullivan, Alam & Matava, 2018*). This emphasizes the significance of tailoring/customizing VR pain treatment to the person to achieve successful intervention (*Pourmand et al., 2018*). Additional investigation is required on various forms of pain, individual reactions, and user encounters in virtual reality settings. The current state of VR pain management applications needs more significant progress, and this issue can only be resolved by prioritizing design in product

development and fostering multidisciplinary cooperation in medical research (*Riva & Serino, 2020*).

While VR has shown potential in the treatment of cancer pain in the middle-aged and elderly, several drawbacks should be considered before using it in a therapeutic context. For example, motion sickness, nausea, and other uncomfortable symptoms are common in middle-aged and elderly adults (*Bani Mohammad & Ahmad, 2019*; *Hundert et al., 2021*). The highly realistic content, virtual animation, and moving virtual scenes in virtual reality settings often exacerbate motion sickness symptoms. As technology progresses, therapists should carefully evaluate the length and extent of VR treatments. Additionally, it is crucial to establish a sense of comfort for participants in the VR environment before the intervention to reduce the likelihood of any negative consequences.

Undoubtedly, virtual reality is a reliable and secure supplement to pain treatment (*Indovina et al., 2018*). However, Hospitals have challenges regarding the viability of adopting and sustaining virtual reality technology. Furthermore, the present availability of VR material that is both practical and effective is restricted, particularly in terms of using non-intrusive techniques. According to *Spiegel (2018)*, younger patients showed more receptiveness towards using virtual reality, whereas middle-aged and elderly patients saw it as invasive, unpleasant, or perplexing. The suggestion was made to establish a "virtual reality pharmacy." Researchers have advocated conducting longitudinal trials with larger sample sizes and verifying the mental states of intervention patients by integrating patient self-reports with physiological parameters (such as stress assessment) to effectively showcase VR in treating pain (*Indovina et al., 2018*). The growing need to investigate novel approaches for improving pain resistance using virtual reality indicates a promising future for this field of study and application. Several factors might enhance the mental well-being of patients. For instance, improving the level of immersion in virtual environments, specifically by incorporating, tailoring, and adapting VR to suit individual variations in preferences and accessibility for diverse demographics (*Pourmand et al., 2018*). Expanding on "customizing the experience," incorporating various sensory modalities in VR interactions can enhance engagement and open up possibilities for VR use among individuals with visual impairments. Nevertheless, to achieve optimal results, significant efforts must be dedicated to improving the empirical foundation supporting the incorporation of VR into conventional analgesic management programs.

## LIMITATION

Our study may improve the physical therapy of people diagnosed with clinical cancer. However, our research still has several possible concerns. RCTs conducted in English may be susceptible to publication bias. Second, the sample size included in this study is limited, and the validity of the results should be carefully explained. Furthermore, the diversity of inpatient rehabilitation programs, chemotherapy regimens, and illness kinds might impact the fluctuation of the combined results in Meta-analysis.

## CONCLUSIONS

This meta-analysis evaluated VR therapies on symptom management in middle-aged and elderly adults with cancer. This research suggests that VR substantially impacts the emotional and physical health of middle-aged and elderly cancer patients. It effectively reduces symptoms of despair, anxiety, and pain. To evaluate the impact of VR on mood regulation and general well-being in middle-aged and elderly cancer patients, following clinical research should incorporate more significant cohorts of participants and prolong the duration of follow-up in randomized controlled trials. Addressing symptoms in cancer patients is a prolonged process.

### Funding

The present study was funded by the Key R&D Plan Projects in Shandong Province (2019GSF108203). The funders had no role in study design, data collection and analysis, decision to publish, or preparation of the manuscript.

### Grant Disclosures

The following grant information was disclosed by the authors:
Key R&D Plan Projects in Shandong Province: 2019GSF108203.

### Competing Interests

The authors report no other conflicts of interest in this work.

### Author Contributions

- Yang Chen conceived and designed the experiments, analyzed the data, prepared figures and/or tables, and approved the final draft.
- Hui Meng conceived and designed the experiments, authored or reviewed drafts of the article, and approved the final draft.
- Qian Chen performed the experiments, prepared figures and/or tables, and approved the final draft.
- Wendong Wu performed the experiments, analyzed the data, prepared figures and/or tables, and approved the final draft.
- HaiBin Liu performed the experiments, analyzed the data, prepared figures and/or tables, and approved the final draft.
- Shi Lv conceived and designed the experiments, authored or reviewed drafts of the article, and approved the final draft.
- Liang Huai conceived and designed the experiments, authored or reviewed drafts of the article, and approved the final draft.

### Data Availability

This is a systematic review and meta-analysis.

## Supplemental Information

Supplemental information for this article can be found online at http://dx.doi.org/10.7717/peerj.18701#supplemental-information.

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
