# Peer review of "Virtual reality therapy in managing cancer pain in middle-aged and elderly: a systematic review and meta-analysis"

_PeerJ, doi:10.7717/peerj.18701_

## Round 0.1 · original submission · Major Revisions

Dear Authors

Even though we have one reviewer (R2) who suggested that we reject your paper we would like to give you an opportunity to address all the concerns raised by the reviewers before making a final decision on your paper. Kindly submit a detailed rebuttal letter responding to the concerns of both the referees in addition to submitting the updated draft of your paper

Reviewer 1 ·

Basic reporting

In this section, the authors comply with all the required items.

Experimental design

In this section, the authors comply with all the required items.

Validity of the findings

In this section, the authors comply with all the required items.

Additional comments

First of all, thank you very much for giving me the opportunity to review this manuscript. I think the subject matter is of great interest. I will now make a proposal for changes to be made to improve the quality of the manuscript:
-It would be useful to update the search as several months have passed and I believe there may be new studies of interest to include.
-The search strategy entered in each database should be explicitly stated so that it can be reproduced by any reader.
-Some reference should be introduced to justify in the PICOS P that the participants included will be aged 60 years or more.
-In the meta-analysis, Cochrane states that choosing a fixed or random effects model should not be done on the basis of heterogeneity.
-In meta-analysis, add the sample size (n=).
-For Figure 2, make it following the latest model provided by PRISMA. Also, specify more about how many studies were found in each database, the exclusion criteria...
-For risk of bias, use the new Cochrane tool, RoB-2.

Reviewer 2 ·

Basic reporting

The manuscript presents significant issues with language quality that hinder its readability and overall comprehension. Many sections contain unclear and grammatically incorrect sentences, making it challenging to follow the narrative and understand the key points being conveyed. Furthermore, numerous technical terms are not used correctly, leading to potential misunderstandings and inaccuracies in the interpretation of the research findings.

Experimental design

This systematic review aims to investigate the effects of virtual reality on cancer pain in older individuals. According to the inclusion criteria, the elderly people (age > 60) with cancer diagnoses were the population of interest. However, only two included studies (Turrado 2021 and Villumsen 2019) met this criterion. The mean ages of the participants in the other five studies were all below 60. The inclusion of these studies clearly contradicts the eligibility criteria. This is a fundamental issue of this systematic review.

Methods - The tool used for risk of bias assessment was outdated (Higgins et at. 2011). The updated ROB-2 should be used (https://doi.org/10.1136/bmj.l4898).

Methods - In statistical analysis, the presentation of the two equations was unclear.

Results - 3.5 Pain in all age groups: "Four studies were conducted with older individuals...and five with adolescents...". How were studies for adolescents included in data analysis? This contradicts not only the eligibility criteria but also the object of this review.

Results - 3.5 Sensitivity analysis: The results of sensitivity analysis were not presented in this manuscript. The credibility is questionable.

Validity of the findings

Please see the main concerns stated above - the validity of this present systematic review and meta-analysis is deemed poor because many included studies violate the eligibility criteria, even though it was stated "two investigators positively assessed the literature to see whether it fulfilled the criteria for inclusion. When the two investigators could not reach a consensus, they spoke with a third investigator (L.H) to decide whether to accept or reject the research". The trustworthiness of the research process is very questionable.

---

## Round 0.2 · accepted · Accept

Dear Authors

Thank you for making all the changes as suggested by the referees. Congratulations on the acceptance of your paper.

Reviewer 1 ·

Basic reporting

-

Experimental design

-

Validity of the findings

-

Additional comments

Dear authors, after re-reading your manuscript, I have seen that all the proposed improvements have been made. The article has gained a lot of quality by updating and improving it and therefore I propose to accept and publish it.
Thank you very much

Reviewer 2 ·

Basic reporting

Concerns addressed.

Experimental design

Concerns addressed.

Validity of the findings

Concerns addressed.

Additional comments

All previous comments have been addressed adequately.